# Research on the Anti-Risk Mechanism of Mask Green Supply Chain from the Perspective of Cooperation between Retailers, Suppliers, and Financial Institutions

**DOI:** 10.3390/ijerph192416744

**Published:** 2022-12-13

**Authors:** Haibo Chen, Zongjun Wang, Xuesong Yu, Qin Zhong

**Affiliations:** 1School of Management, Huazhong University of Science and Technology, Wuhan 430074, China; 2School of Economics and Management, Hubei Polytechnic University, Huangshi 435003, China; 3School of Management, Wuhan Polytechnic University, Wuhan 430048, China

**Keywords:** mask, green supply chain, anti-risk, pandemic

## Abstract

Against the background of the pandemic, the mask supply chain faces the risk of pollution caused by discarded masks, the risk of insufficient funds of retailers, and the risk of mask overstock. To better guard against the above risks, this study constructed a two-party game model and a cusp catastrophe model from the perspective of the mask green supply chain, and studied the strategic choices of retailers and suppliers in the supply chain affected by the risk of capital constraints and overstock. The result shows that the risk shocks will lead to the disruption of the mask green supply chain, and the main factors affecting the strategy choice of mask suppliers and retailers are mask recycling rate, deposit ratio, risk occurrence time, etc. In further research, this study involved a mechanism for financial institutions, mask retailers, and the government to jointly deal with the risk of mask overstock, the risk of retailers’ insufficient funds, and the risk of environmental pollution from discarded masks. The research path and conclusion of this study reveal the risks in the circulation area of mask supplies during the pandemic, and provide recommendations for planning for future crises and risk prevention.

## 1. Introduction

The coronavirus disease 2019 (COVID-19) pandemic shows that it is necessary to improve the ability of the supply chain to avoid risks during the pandemic so that it can quickly return to a stable state after being damaged and avoid supply chain disruption [1]. As important personal protective equipment (PPE) [2], during the COVID-19 pandemic, masks were considered by governments and public health experts to help prevent the spread of viruses in the community and protect public health [3]. Therefore, the consumption of masks increased rapidly during the COVID-19 pandemic, resulting in a worldwide shortage of masks [4]. In order to solve the shortage of masks and resist the COVID-19 pandemic, retailers in various countries have increased the purchase of masks to meet consumers’ demand for masks. Together with suppliers in mask production, these retailers form a mask supply chain, which is an important part of the PPE supply chain and an important part of the emergency supply chain during the pandemic [5,6]. During the COVID-19 pandemic, the rising consumption of masks also brought corresponding environmental pollution problems. A study from University College London’s Plastic Waste Innovation Hub has cited a single-use face mask each day for a year would generate 66,000 tonnes of contaminated plastic waste [7]. However, the uncertainty and periodicity of the COVID-19 pandemic led to the vulnerability or even disruption of the mask supply chain due to the sharp increase or drop in mask sales. Therefore, it is necessary to improve the green level and resilience of the mask supply chain to reduce the environmental pollution in the process of mask supply and reduce the risk shocks in the mask supply chain [8].

The legal constraints related to environmental protection and the public pressure to pursue greening have made the suppliers bear the responsibility for environmental protection. In order to reduce and pass on the high environmental protection costs, suppliers tend to cooperate with retailers and share environmental protection responsibilities and costs [9]. Therefore, if mask suppliers and retailers choose to share the responsibility of environmental protection and join the green supply chain, they can reduce the risk of environmental pollution caused by the rapidly growing discarded masks because of the pandemic, avoid the risk of being punished by the government with environmental protection-related laws, and avoid the potential risk of affecting their public image due to the criticism of environmental protection organizations and public opinion.

Improving the green level of the mask supply chain will increase the green cost of mask retailers and suppliers. In addition, the uncertainty and periodicity of the pandemic lead to the risks of overcapacity and overstock of the mask supply chain, which may lead to the disruption of the mask supply chain due to high operating costs, bringing loss of benefits to mask suppliers and retailers [10]. This is because when the pandemic ends or subsides in stages, the market demand for masks may return to the level before the pandemic, resulting in a sharp drop in mask sales. As the mask sales contract between the mask supplier and the retailer generally stipulates that the retailer will obtain the masks after paying the deposit to the supplier and sell them to the consumer, and then pay the remaining payment after the masks are sold out, if the masks are unsalable during the sales process, the retailer will overstock a large number of masks and be unable to make the payment to the supplier, resulting in the receivable on the masks by the supplier not being received. Eventually, the mask retailers and suppliers suffer economic losses due to the risk of overstock and overcapacity, respectively. In order to avoid the risk of overcapacity, some mask suppliers required mask retailers to adopt the full order model instead of the traditional deposit order model during the COVID-19 pandemic [11]. However, because most mask retailers are small and medium-sized retailers, they lack sufficient funds to accept the full order model, which leads to the incoordination between mask suppliers and retailers, directly affecting the sustainable operation of the mask green supply chain and even leading to supply chain disruption [12]. For example, the COVID-19 pandemic in mainland China has been effectively controlled since April 2020, resulting in a decline in demand, overstock, and a sharp drop in prices in the Chinese mask market after May 2020. To cope with the excess demand for masks, some mask suppliers were forced to sell mask production equipment and raw materials at low prices, while some mask retailers stopped selling masks, thus affecting the sustainable operation of the mask supply chain [13]. Therefore, mask suppliers and retailers should cooperate to establish an anti-risk mechanism for the mask supply chain against the background of the pandemic, effectively resolve the risks faced by mask suppliers and retailers, and ensure that the mask supply chain can still maintain sustainable operation and safeguard its own interests under the risk shocks.

Against the background of the pandemic, the mask supply chain faces the risk of pollution caused by discarded masks, the risk of insufficient funds of retailers, and the risk of mask overstock. The literature mainly studies mask anti-risk from the perspective of suppliers by analyzing mask production, mask price, and financial subsidies. However, the pollution and recycling of discarded masks cannot be separated from the close cooperation between retailers and suppliers in the mask supply chain [14], and the research methods based on suppliers’ perspectives struggle to solve the risks of retailers’ insufficient funds and mask overstock. In order to overcome the shortcomings of existing research, the anti-risk of the mask supply chain against the background of the pandemic was studied from the perspective of cooperation among mask retailers, suppliers, and financial institutions.

The study is mainly divided into three stages: The first stage designs the basic process of the mask green supply chain from the perspective of cooperation between retailers and suppliers. Based on the literature review and practice summary, the mechanism of upgrading the traditional mask supply chain to a green supply chain is preliminarily established to prevent the environmental pollution risk of discarded masks; the second stage analyzes the evolution law of the mask supply chain under the risk shocks of mask overstock against the background of the pandemic. A two-party game model between mask retailers and suppliers is established to analyze the change in strategy selection between retailers and suppliers under the risk shocks of mask overstock, and then the cusp catastrophe model is used to analyze the process and conditions of the change in strategy selection between retailers and suppliers; the third stage establishes an anti-risk mechanism in which the government, financial institutions, mask retailers, and suppliers work together against the background of the pandemic. First, through a questionnaire survey of employees in the financial industry, we can understand their attitude toward using factoring as a financial instrument to finance the mask green supply chain. Then, a tripartite game model among mask retailers, factoring institutions, and the government is established, and the model is solved and tested. On this basis, the cooperation mechanism of the government, financial institutions, mask retailers, and suppliers to jointly prevent risks against the background of the pandemic is analyzed. The goal of this study is to explore the anti-risk mechanism of the mask supply chain so as to improve the green level and resilience of the mask supply chain against the background of the pandemic and meet the capital demand of the mask supply chain and ensure the sustainable operation of the mask supply chain.

## 2. Literature Review

### 2.1. Mask Green Supply Chain

Disposable medical and civil masks are frequently used by ordinary consumers. This kind of mask is mainly composed of a waterproof layer (spunbond non-woven fabric), filter layer (meltblown non-woven fabric), ear band, nose bridge strip, and other components. Among them, the waterproof layer and filter layer are made of polypropylene non-woven superfine fiber [15], which belongs to the downstream products of the petrochemical industry and easily becomes plastic particles in the environment [16]. Compared with smog particles, plastic particles are more harmful because plastic particles are difficult to degrade in the environment. They can enter animal and even human bodies through water and food, which seriously threatens the health of people and animals and destroys terrestrial and marine ecosystems [17]. If discarded masks are burned or buried as garbage, it will not only bring secondary pollution but also waste mask raw material resources [18,19]. The industrial and scientific circles put forward processes to improve masks to prolong the service life of masks [20]; they are to use green fibers in mask raw materials to reduce environmental pollution [21], use recyclable biomaterials to produce reusable masks [22], and use new pyrolysis technology to extract and recycle petrochemical raw materials in discarded masks [23].

Based on studying the new disposal technology of discarded masks, some enterprises put forward the idea of building a green supply chain of masks and establishing a perfect mask recycling system and putting it into practice. For example, in March 2022, Fraunhofer Institute for Environmental, Safety and Energy Technology (UMICSHT, Oberhausen, Germany), Sabic Innovative Polymers (SABIC), and Procter & Gamble (P&G, Cincinnati, OH, USA) announced a collaboration of an innovative circular economy pilot project aimed at proving the feasibility of closed-loop recycling of disposable masks. The basic process of this project is to set up a waste mask collection box at P&G to collect the waste masks of employees, then transport them to UMICSHT, convert them into cracked oil using a chemical method, and then send them to SABIC for production of new polypropylene resin, which is the main raw material of masks [24]. “Green” in the green supply chain involves the relationship between supply chain management and the natural environment. Against the background of the pandemic, the industrial chain should become resilient and have sustainable development ability, thus reducing the negative impact on the natural environment [25]. Establishing a green supply chain through green cooperation among enterprises is not only conducive to their own sustainable development but also conducive to obtaining green subsidies and tax incentives from the government, obtaining reputation capital, and enhancing the brand loyalty of consumers [26]. Some researchers have designed a sustainable green supply chain network of the masks during the pandemic, which solves the decision-making problems of site selection, supply, production, distribution, collection, isolation, recycling, reuse, and disposal in the supply chain [27].

### 2.2. Supply Chain Resilience and Anti-Risk

Supply chain resilience is the adaptive ability and reparative ability of the supply chain to cope with shocks, recover quickly or reach other better states after being impacted by risks, avoid supply chain disruption, and ensure the steady operation of the supply chain [28]. When a risk event occurs, enterprises are able to cope with the shocks caused by the risk event by timely integrating and configuring the internal and external resources in the supply chain network and improving the operational efficiency of the supply chain [29]. The supply chain with strong resilience has the ability to better organize internal resources, processes, and systems when it faces uncertain factors. It can predict and reduce the negative impact caused by the uncertainty of the pandemic [30]. Facing the risk of supply chain disruption, enterprises can improve supply chain resilience by developing an anti-risk mechanism, overcoming the threat of supply chain disruption and enabling them to continuously provide goods and services to customers [31]. If the retail supply chain lacks resilience, there is a risk of disruption. Therefore, an anti-risk mechanism for the retail supply chain should be established to improve the resilience and robustness of the supply chain to reduce the risk of disruption [32]. Emergencies may lead to the disruption of the green emergency supply chain and hinder the supply chain from playing its role in reducing environmental pollution and saving resources [33].

Scarcity of labor, shortage of raw materials, and inconsistent supply are three major challenges faced by the global supply chain during the COVID-19 pandemic [34]. Deep learning can help decision makers actively predict supply chain risks against the background of the pandemic and improve the supply chain’s resilience [35]. Strategies such as continuous monitoring, information sharing, and real-time data exchange are helpful in dealing with external risks in the supply chain [36]. During the pandemic, governments of various countries jointly purchase essential drugs and supplies urgently needed by patients, minimizing the risk of medical supply chain disruption and alleviating the shortage of medical materials [37]. The WHO, governments, and medical suppliers should jointly examine the weaknesses and failures of the medical supply chain during the pandemic so as to facilitate the establishment of a new medical and health system [38]. Adopting a multi-channel supplier sourcing strategy, strengthening communication between upstream and downstream of the mask supply chain, and close cooperation with freight forwarders have a positive effect on preventing and mitigating mask supply chain risks during the pandemic [12].

### 2.3. Financing of Mask Supply Chain

To improve the supply chain resilience against the background of pandemic, it is necessary to increase the supply of resources, including funds. Therefore, supply chain finance should be introduced to establish a resource system to ensure the operation of the supply chain [39]. Against the background of the pandemic, if the government can provide financial subsidies to small and medium-sized enterprises, it will help reduce the financing cost of the latter and the financing risks of financial institutions [40]. When the market demand of the green supply chain is uncertain, suppliers tend to require downstream customers of the supply chain to pay in advance for all goods to solve the cost constraint problem when suppliers improve the green level of products [41]. In order to ensure the sustainable operation of the PPE supply chain, including masks, financial institutions should be coordinated with the support of the government to provide financing services for enterprises in the PPE supply chain so as to expand PPE supply and meet people’s demand for PPE such as masks during the pandemic [42].

Factoring is a financial instrument that is suitable for retailers, and it helps to improve the overall performance of the supply chain and avoid supply chain disruption due to financial problems [43]. Factoring has many financing models, among which the reverse factoring model makes full use of the core position of core enterprises in the supply chain and provides financing to suppliers with the credit of core enterprises as a guarantee [44]. When retailers have insufficient self-owned funds, they can strengthen cooperation with upstream suppliers, share the credit of the latter, and obtain financing from financing institutions [45].

## 3. Model Construction

### 3.1. Two-Party Game Model of the Anti-Risk Capability of Mask Green Supply Chains

According to the concept of the green supply chain, the mask supply chain should have the ability to recycle and dispose of discarded masks to reduce the risk of environmental pollution. According to this concept, in the green supply chain of masks designed in this study, mask suppliers should sell masks to mask retailers in batches according to a certain time cycle, and retailers should set up recycling points for discarded masks while selling masks to consumers so that consumers could place used masks in the recycling points. At the end of this time cycle, the mask retailer is responsible for returning the collected discarded masks to the mask supplier, who will process the discarded masks and obtain the raw materials for mask production, such as polypropylene in the adhesive-bonded fabric of masks. If the masks are unsalable in this cycle, the mask retailer will return the overstocked and discarded masks to the mask supplier.

Considering the strong impact and uncertainty of the pandemic, the mask supply chain faces external risks, which will make mask suppliers and retailers less willing to accept the recycling scheme of discarded masks for fear of economic losses. Therefore, this study constructed a two-party game model of the anti-risk capability of the mask green supply chains with the mask supplier and retailer as the main actors. Through model derivation, analyses on whether the basic process of the mask green supply chains initially designed in this study could remain stable under the impact of external risks were conducted.

The assumptions of the two-party game model designed in this study are as follows:

**Assumption** **1.**
*During the pandemic, the mask supplier will sign a mask sales and recovery contract with the retailer and will sell masks in N phases to the latter; the number of masks sold in each phase is M. The cost of producing masks by suppliers is P_1_/piece, the wholesale price sold to retailers is P_2_/piece, and the sales price sold by retailers to consumers is P_3_/piece.*


**Assumption** **2.**
*During the pandemic, there are two ordering methods for mask suppliers and retailers: One method is deposit order, that is, the mask retailers are allowed to obtain masks by paying a certain proportion of the purchase price as a deposit, and the remaining payment will be paid after the end of the cycle. The ratio of the deposit to the total purchase price is λ; the other method is full order, that is, the mask supplier will deliver the masks to the retailer after obtaining the full payment for the masks in this cycle; under this method, the retailer needs to prepare an additional C4 payment to avoid insufficient funds for payment.*


**Assumption** **3.**
*In the process of cooperation between the mask supplier and the retailer, the use and maintenance cost of the mask recycling equipment is C1/cycle, which is borne by the mask supplier; mask retailers set up discarded mask facilities to recycle discarded masks from consumers at C3/cycle and sell them to mask suppliers at P4/piece. The proportion of the number of discarded masks recycled by mask retailers to the number of masks sold to consumers in the current period is δ, and the ratio of the revenue from recycling discarded masks by mask suppliers to the cost of mask manufacturing is β.*


**Assumption** **4.**
*In the t (t < n) phase of the contract between the mask supplier and the retailer, the masks will be impacted by the risk of unsalability. The ratio of unsalable masks to masks produced in each phase is ε. If the mask retailer chooses to terminate the contract and stop cooperation after the masks are unsalable, the penalty to be paid to the mask supplier is D. After the early termination of the cooperation between the two parties, the mask retailer does not need to pay the remaining payment of the current period. The current mask inventory cost of the mask retailers is C2.*


**Assumption** **5.***The set of game strategies of mask suppliers is {deposit order, full order}, and the set of game strategies of mask retailers is {maintain cooperation, terminate cooperation}, where the probability of mask suppliers choosing deposit order and the full order is k and 1-k, respectively, and the probability of mask retailers choosing to maintain cooperation and terminate cooperation is s and 1-s, respectively. A “0” means that the mask supplier chooses to place a full order, and the mask retailer chooses to stop cooperation, “1” means that the mask supplier chooses to place a deposit order, and the mask retailer chooses to maintain cooperation*.

Based on the above assumptions, this study constructs a two-party game model for the anti-risk capability of the mask green supply chains. The income matrix of the model is shown in Table 1:

Table 1 shows that the dynamic replication equation of the behavior of mask retailers choosing to continue to cooperate with mask suppliers is:G(s) = s(1 − s){s[m(t − 1)(−P_2_) + m(1 − ε)(n − t + 1)(P_3_ − P_2_) − (n − t + 1)C_2_] + (1 − s)[m(t − 1)(P_3_ − P_2_) − λmP_2_ + (1 − ε)mP_3_ − C_2_ − D] − s[m(t − 1)(P_3_ − P_2_) + m(1 − ε)(n − t + 1)(P_3_ − P_2_) − (n − t + 1)C_2_ − f] − (1 − s)[m(t − 1)(P_3_ − P_2_) − mP_2_ + (1 − ε)mP_3_ − C_2_ − D]}(1)
         Let A = m(t − 1)(P_3_ − P_2_) + m(1 − ε)(n − t + 1)(P_3_ − P_2_) − (n − t + 1)C_2_, B = m(t − 1)(P_3_ − P_2_)-λmP_2_ + (1 − ε)mP_3_ − C_2_ − D,            E = m(t − 1)(P_3_ − P_2_) + m(1 − ε)(n − t + 1)(P_3_ − P_2_) − (n − t + 1)C_2_ − f,  F = m(t − 1)(P_3_ − P_2_) − mP_2_ + (1 − ε)mP_3_ − C_2_-D,

At this time, the copied dynamic equation can be converted to
G(s) = −(A − B − E + F)s^3^ + (A − 2B − E + 2F)s^2^ + (B − F)s(2)

In this study, we calculated that when m = 100, n = 100, t = 10, C_1_ = 10, C_2_ = 10, C_3_ = 10, C_4_ = 10, P_1_ = 9, P_2_ = 9.7, P_3_ = 100, P_4_ = 100, D = 0.01, δ = 0.036, λ = 0.1, ε = 0.9, the equilibrium point of the two-party game model is (0, 0), which means that under the pressure of external risks, the mask supplier has chosen to require the retailer to place a full order. The mask retailer has chosen to terminate cooperation with the supplier, which means that the mask green supply chain initially designed in this study against the background of the COVID-19 pandemic cannot operate normally under the impact of risks and cannot ensure that consumers can obtain enough masks during the pandemic. It is also impossible to ensure the timely recycling and disposal of discarded masks.

### 3.2. The Cusp Catastrophe Model of the Anti-Risk Capability of the Mask Green Supply Chain

To better observe and analyze the critical conditions and the evolution process of the mask green supply chain’s interruption or resumption of normal operation under the impact of risk in a disease pandemic, this study introduced a cusp catastrophe model to describe them. The basic principle of the cusp catastrophe model is that the change in the two control variables will cause a change in the state variables of the system. As shown in Figure 1, the three-dimensional surface of the cusp catastrophe model is composed of upper leaves, lower leaves, and pleated leaves. The upper leaves and lower leaves are safety layers, and the pleated leaves are risk layers. O is the starting point of system evolution. When O evolves from the upper lobe to the lower lobe along the O–S path, the system is always in a safe state; however, when the system evolves along the O–P–H–Z path, it passes through the folded leaves in the P–H interval. At this time, the system suddenly changes into a risk state and the operation trajectory becomes unstable.

In this study, the strategy that the mask retailer chooses to continue to cooperate with the mask supplier under external risk pressure is taken as the security state in the cusp catastrophe model, the strategy that the mask retailer chooses to stop cooperating with the mask supplier under external risk pressure is taken as the risk state in the cusp catastrophe model. The dynamic replication equation of the selection behavior of the mask retailer in the two-party game model between the mask supplier and the retailer is translated and transformed into the relevant equations, parameters, and constraints of the cusp catastrophe model under corresponding conditions which then analyze the evolution process and conditions of the attitude of mask retailers towards the mask green supply chain when facing external risks through the catastrophe model. Refer to the formula derivation process of relevant papers [46,47], and the process of translation and transformation of this paper is as follows:Let N_1_ = −(A − B − E + F),N_2_ = A − 2B − E + 2F, N_3_ = B − F, then N_2_ = −(N_1_ + N_3_)

Formula (1) can be converted to
(3)$$G(s)=\frac{dm_{1}}{dt}=N_{1}{m_{1}}^{3}+N_{2}{m_{1}}^{2}+N_{3}m_{1}$$
(4)$$\mathrm{Let}\;y=m_{1}+\frac{N_{2}}{3N_{1}},\mathrm{then}\;\frac{dy}{dt}=y^{3}(t)+\mathrm{uy}(t)+v$$

Equation (3) can be changed to the balance surface equation of the standard cusp catastrophe model as
V′ = y^3^ + uy + v = 0(5)

Among them, “u” and “v” are the control variables, while “y” is the state variable. The u and v variables are calculated as follows:(6)$$u=\frac{N_{3}}{N_{1}}-\frac{N_{2}^{2}}{3N_{1}^{2}}=\frac{3N_{1}N_{3}-N_{2}^{2}}{3N_{1}^{2}}\;,\;v=\frac{N_{2}^{3}}{27N_{1}^{3}}-\frac{N_{2}N_{3}}{3N_{1}^{2}}$$

According to the catastrophe theory, the critical surface where the risk state occurs is the projection of the curved surface, as shown in Equation (5), in the direction (u, v). Taking the derivative of Equation (4), the condition that the singular point set satisfies is
V″ = 3y^2^ + u = 0(7)

Simultaneously, Equations (4) and (6) can obtain the critical surface of the patient’s behavior in the risk state; that is, the set of divergence points is computed as:4u^3^ = −27v^2^(8)

As shown in Figure 2, when the value of v remains unchanged and the value of u gradually increases, the phase point motion track is B_2_–S_2_, which will not cause sudden changes in the mask supplier status and will continue to be in a safe state. When the value of v gradually decreases and is less than 0, and the value of u remains > 0, the phase point trajectory changes to S_2_–A_2_–D_2_ and the phase point moves from upper leaf to lower leaf along the smooth surface. At this time, there is no intersection with the set of bifurcation points on the two-dimensional plane $B_{2}^{\prime }$–$S_{2}^{\prime }$–$A_{2}^{\prime }$–$D_{2}^{\prime }$, and the behavior of the mask retailer is always in a safe state during the evolution process.

When u > 0, Equation (6) has no real number solution; that is, the changed potential function has no real number solution, and the risk state will not appear at this time according to Equation (5), N_1_ > 0 and N_3_ > 0. According to the definition of each parameter in the two-party game model, it can be seen that the mask retailer will choose the strategy of full order at this time as λ < 1, then λ MP_2_ < mP_2_ is always established, and the behavior of mask retailers is always in a safe state, that is, mask retailers always choose the full order mode.

As shown in Figure 3, when the value of u remains < 0 and the value of v increases, the phase point motion track is L_1_–L_2_, and the mask supplier/retailer is in a safe state; with the value of v further increasing, the phase point motion track turns to L_2_–L_3_. At this time, the phase point is close to the pleated leaf, and the mask supplier/retailer’s behavior gradually changes from a safe state to a risky state and, with the further increase in v value and the keeping of u value < 0, the phase point motion track changes to L_3_–L_4_. At this time, the phase point enters the pleated leaf and jumps suddenly. This is shown in the two-dimensional plane as the projection of the phase point motion track crosses the set of divergent points $L_{4}^{\prime }O^{\prime }L_{7}^{\prime }$. At this time, the behavior of the mask supplier/retailer changes suddenly. In the pleated leaf, the phase point is very unstable and there is no fixed motion track. As the v value further decreases to a positive value, the phase point motion track turns to L_5_–L_6_, and the phase point gradually leaves the pleated leaf and enters the lower leaf. The mask supplier/retailer’s behavior changes from a sudden state to a safe state.

When u < 0, Equation (6) has a real number solution. There is a possibility of mutation and entering the risk state in the bifurcation point set at this time, 3N_1_^2^ < N_2_^2^. U < 0 can be achieved by increasing the value of N_2_ and decreasing the value of N_1_ and N_3_. At this time, the difference from A–E increases. According to Table 1, n-t decreases, C decreases and M increases, ε reduces, and λ and D_2_ increase, which means that the time of risk occurrence and the contract expiration are shortened, the number of masks in overstock is reduced, the number of masks ordered at one time is increased, the proportion of deposit in payment for goods is increased, and the proportion of unsalable masks in the number of masks purchased in the current period is reduced. The probability of mask retailers continuing to provide services is increased.

After transforming the dynamic replication equation of the mask supplier, it is found that n-t decreases, C decreases, m increases, ε reduces, and D_1_ increase, means that the time of risk occurrence and the expiration of the contract are shortened, the number of masks overstocked in inventory is reduced, the number of masks ordered at a time is increased, the proportion of deposit in payment is increased, and the efficiency of recovering masks is improved, so the probability of mask suppliers to continue to provide services is increased.

## 4. Further Research Introducing Financial Institutions into the Green Supply Chain of Masks

### 4.1. Questionnaire

As described in the Section 1 of this paper, factoring is a financial instrument suitable for providing financing to retailers in the supply chain. In order to fully understand the attitude of financial institutions towards the use of factoring businesses to support the green supply chain of masks in the context of the pandemic, we designed a related questionnaire and distributed it online to the practitioners of factoring businesses in financial institutions. A total of 54 online questionnaires were distributed, including 54 valid ones.The items of the questionnaire are in the Appendix B.

As for the job description of the respondents in the questionnaire, Figure 4 shows that 22.22% of the respondents are senior executives of financial institutions, 53.7% are general managers of market departments in financial institutions, and 18.52% are general managers of products, risk control, or operation departments. In addition, 40.74% of respondents have worked in the factoring field for more than 5 years, and 27.78% have worked in the factoring field for 2–5 years. Using the 5-point system, the respondents’ understanding of the production and marketing of masks and the recovery technology of masks scored 2.41 and 2.48, respectively (“1” means “completely unknown,” “5” means “completely known”).

Regarding the risks faced by financial institutions in providing financial services to the mask green supply chains, 72.22% of the respondents were concerned about the risk of default of mask suppliers and retailers, and 64.81% of the respondents were concerned that mask recycling technology was just a tool used by suppliers to defraud money. Of the respondents, 75.93% believed that if financial institutions could supervise the process of recycling and utilization of discarded masks, they would increase their willingness to provide financing for the green supply chain of masks. Additionally, 64.81% of the respondents took a positive attitude towards the risk-sharing mechanism of applying blockchain technology to mask green supply chains. The detailed data of answer sheet is in the Appendix A of this paper.

### 4.2. Tripartite Game Model of Respirator Retailers, Suppliers, and Government

As shown in the Section 2 of this paper, under the impact of external risks, the mask green supply chain faces the risk of disruption due to the lack of resilience. In order to improve the resilience of the green supply chain of masks, it is necessary to establish and improve the risk prevention mechanism of the supply chain of masks, improve the financial constraints of retailers, and eliminate the risk of overstocking masks. Based on the above analysis and questionnaire results, we constructed a tripartite game model based on the risk prevention mechanism of the mask green supply chain of factoring financial institutions, mask retailers, and the government to improve the resilience of the mask supply chain and better resist risks. The model assumptions are as follows:

**H1.** *During the pandemic period, mask suppliers only accept full payments ordered by retailers*.

**H2.** 
*Mask retailers use the reverse factoring model in factoring business to finance, sign a third-party factoring contract with financial institutions and mask suppliers engaged in factoring business, pay interest in n phases according to the contract, and pay off the principal in a lump sum at maturity. The transfer of accounts receivable under the factoring contract has no recourse against the supplier and has recourse against the retailer. In phase t, masks were unsalable and overstocked.*


**H3.** 
*The government’s strategy is {support, not support}, the financial institution’s strategy is {maintain, early withdrawal}, and the mask retailer’s strategy is {maintain, default}. Among them, the probabilities of factoring financial institutions choosing “maintenance” and “early recovery” strategies are m and 1 − m, respectively. The probabilities of mask retailers choosing “maintenance” and “default” strategies are s and 1 − s, respectively, and the probabilities of governments choosing “support” and “not support” are x and 1 − x, respectively.*


**H4.** *Factoring financial institutions need to pay liquidated damages Y if they choose to recover in advance and cannot receive government subsidies. If the mask retailer chooses to default, it will not receive subsidies from the government, and it needs to pay a penalty of W4 (the penalty includes the interest during the default period and the penalty interest). Financial institutions need to pay the collection cost of W3, and the probability of successful recovery is P. If the government chooses to support, the revenue from collecting and storing masks in inventory will be provided to financial institutions. If both the factoring financial institution and the mask retailer voluntarily rescind the factoring contract, there is no need to pay liquidated damages, and neither party can obtain government subsidies*.

**H5.** 
*There are two ways for the government to support the green supply chain of masks. One is to provide financial subsidies to retailers and financial institutions, respectively, to encourage cooperation between the two parties. The amount of subsidies is F1 and F2, respectively, which will be paid after the factoring contract is completed. Second, the government will purchase the overstocked masks from retailers at the market price after the masks become unsalable and store them. The social benefits brought by the government through the above measures to improve the resilience of the emergency supply chain and realize the sustainable operation of the green supply chain of masks are R.*


Based on the above assumptions, the Revenue Matrix of Tripartite Game Mode is built in this paper, as shown in Table 2:

We calculated that when n = 50, W_1_ = 60, W_2_ = 50, W_3_ = 50, W_4_ = 90, F_1_ = 10, F_2_ = 10, U = 0.1, E = 50, R = 90, t = 50, p = 0.9, r = 0.01, the equilibrium point of the entire tripartite game model is (1, 1, 1), representing that financial institutions and mask retailers are willing to maintain cooperation, and the government continues to support the mask supply chain. At this time, the mask green supply chain maintains sustainable operation.
$$\begin{array}{cc} K\left(m\right)=\frac{dm}{dt}=m & \left(1-m\right)\{[\mathrm{sx}(V\left(1-\lambda \right){\left(1+r\right)}^{n}-V\left(1-\lambda \right)-W_{1}-W_{2}+F_{1})+\left(1-s\right)x(V\left(1-\lambda \right){\left(1+r\right)}^{n-t}\\ & -V\left(1-\lambda \right)-W_{1}-W_{2}+F_{1}-W_{3}-\left(1-p\right)V\left(1-\lambda \right)+p\left[V\left(1-\lambda \right)+W_{4}\right]+\mathrm{VU})+s(1\\ & -x)\left(V\left(1-\lambda \right){\left(1+r\right)}^{n}-V\left(1-\lambda \right)-W_{1}-W_{2}\right)+\left(1-s\right)\left(1-x\right)(V\left(1-\lambda \right){\left(1+r\right)}^{n-t}-V(1\\ & -\lambda )-W_{1}-W_{2}-W_{3}-\left(1-p\right)V\left(1-\lambda \right)+p\left[V\left(1-\lambda \right)+W_{4}\right])]-[\mathrm{sx}(V\left(1-\lambda \right){\left(1+r\right)}^{n-t}\\ & -V\left(1-\lambda \right)-W_{1}-W_{2}-Y)+\left(1-s\right)x\left(V\left(1-\lambda \right){\left(1+r\right)}^{n-t}-v\left(1-\lambda \right)-W_{1}-W_{2}\right)+s(1\\ & -x)\left(V\left(1-\lambda \right){\left(1+r\right)}^{n-t}-V\left(1-\lambda \right)-W_{1}-W_{2}-Y\right)+\left(1-s\right)\left(1-x\right)(V\left(1-\lambda \right){\left(1+r\right)}^{n-t}\\ & -V\left(1-\lambda \right)-W_{1}-W_{2})]\} \end{array}$$
$$\begin{array}{cc} H\left(S\right)=\frac{ds}{dt}=s(1 & -s)\{[\mathrm{mx}\left(V\left(1-U\right)\left(1+E\right)-V\left(1-\lambda \right){\left(1+r\right)}^{n}-V\lambda +F_{2}+\mathrm{VU}\right)+m\left(1-x\right)(V\left(1-U\right)(1+E)\\ & -V\left(1-\lambda \right){\left(1+r\right)}^{n}-V\lambda )+\left(1-m\right)x(V\left(1-U\right)\left(1+E\right)-V\left(1-\lambda \right){\left(1+r\right)}^{n-t}-V\lambda +F_{2}+\mathrm{VU}\\ & +Y)+\left(1-m\right)\left(1-x\right)\left(V\left(1-U\right)\left(1+E\right)-V\left(1-\lambda \right){\left(1+r\right)}^{n-t}-V\lambda +Y\right)]-[\mathrm{mx}(V(1-U)(1\\ & +E)-V\left(1-\lambda \right){\left(1+r\right)}^{n-t}-V\lambda +\left(1-p\right)V\left(1-\lambda \right)-p\left[V\left(1-\lambda \right)+W_{4}\right])+m(1-x)(V(1\\ & -U)\left(1+E\right)-V\left(1-\lambda \right){\left(1+r\right)}^{n-t}-V\lambda +\left(1-p\right)V\left(1-\lambda \right)-p\left[V\left(1-\lambda \right)+W_{4}\right])+(1\\ & -m)x\left(V\left(1-U\right)\left(1+E\right)-V\left(1-\lambda \right){\left(1+r\right)}^{n-t}-V\lambda +\mathrm{VU}\right)+\left(1-m\right)\left(1-x\right)(V\left(1-U\right)(1+E)\\ & -V\left(1-\lambda \right){\left(1+r\right)}^{n-t}-V\lambda )]\} \end{array}$$
$$\begin{array}{cc} G\left(x\right)=\frac{dx}{dt}=x(1 & -x)[\mathrm{ms}\left(R-\mathrm{VU}-F_{1}-F_{2}\right)+m\left(1-s\right)\left(R-\mathrm{VU}-F_{1}\right)+(1-m)s\left(R-\mathrm{VU}-F_{2}\right)+(1\\ & -m)\left(1-s\right)\left(R-\mathrm{VU}\right)] \end{array}$$


### 4.3. Tripartite Game Model Test

In order to better understand the evolution process of the three-party game model of the risk prevention mechanism of the mask green supply chain, we used a programming language to draw a simulation diagram of the evolution of the running track of each parameter with time in the tripartite game process, as shown in Figure 5.

It can be seen from the above figure that the running track of each parameter in the tripartite game model eventually converges to (1,1,1), which is consistent with the equilibrium point of the tripartite game model calculated in this study.

In order to further test the robustness of the tripartite game model of the risk prevention mechanism of the mask green supply chain, this study used a programming language to analyze the sensitivity of some parameters in the tripartite game model. The specific method is to expand and reduce the values of these parameters by 30%, respectively. The results are shown in the following Figure 6, Figure 7, Figure 8, Figure 9 and Figure 10.

It can be seen from the above figures that after the values of several parameters of the tripartite game model are changed, their operation tracks converge to (1,1,1), indicating that the model has good robustness.

## 5. Conclusions

The sudden and uncertain nature of the pandemic has exposed the mask supply chain to risks such as overcapacity and overstock. Once the risk breaks out, the mask supplier may require the mask retailer to place a full order to avoid the risk and pass the risk on to the latter, which will lead to the interruption of cooperation between the mask supplier and the retailer, making the mask retailer unable to supply masks to the public and recycle discarded masks, finally interrupting the supply chain of masks.

If the mask retailers, in the context of the pandemic, need to prevent the risk of interrupting cooperation with mask suppliers, they can reduce the supply time and supply batches of masks signed for with the supply chain, reduce the total number of masks ordered under the contract, increase the number of masks ordered in each batch, increase the proportion of deposit in the total payment for goods, take measures to attract more people to give their discarded masks, etc. In order to better prevent the possible risks in the context of a pandemic, mask retailers should cooperate with financial institutions and the government, sign financing contracts with financial institutions under government subsidies, and use reverse factoring, a financial tool, to obtain financing from financial institutions, so that the government and financial institutions can help share and resolve the possible risks and avoid conflicts with suppliers after the risks occur, improving the resilience of the mask supply chain.

## 6. Discussion

The mask supply chain constructed in this study against the background of the epidemic has the characteristics of green recycling, which is based on the pilot use of mask recycling technology organized by P&G, aiming to meet the demand of people for masks during the pandemic and reduce the environmental pollution and resource waste caused by a large number of discarded masks, which conforms to the cutting-edge concept of the green supply chain. The mask recycling process designed in this study is centered on the mask retailers, who are not only responsible for selling masks to consumers but also responsible for recycling discarded masks from consumers and handing them over to the mask suppliers for technical treatment to avoid pollution of toxic chemicals in discarded masks in the environment and impact on human health and to recycle the raw materials for mask production and invest in new mask production links to save the cost of mask production.

## Figures and Tables

**Figure 1 ijerph-19-16744-f001:**
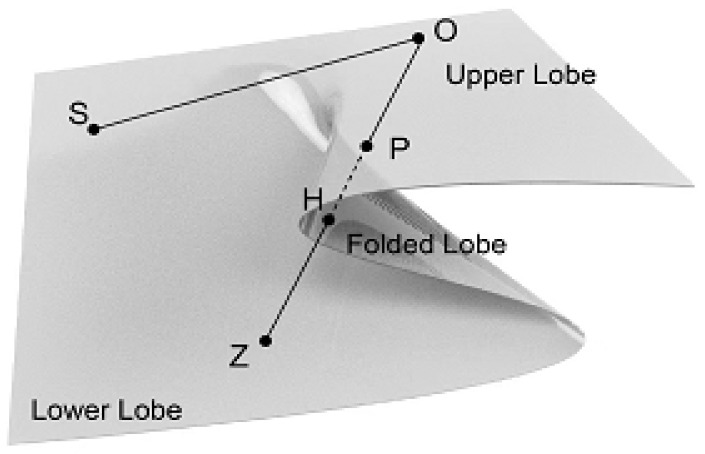
Three-dimensional surface of cusp catastrophe model.

**Figure 2 ijerph-19-16744-f002:**
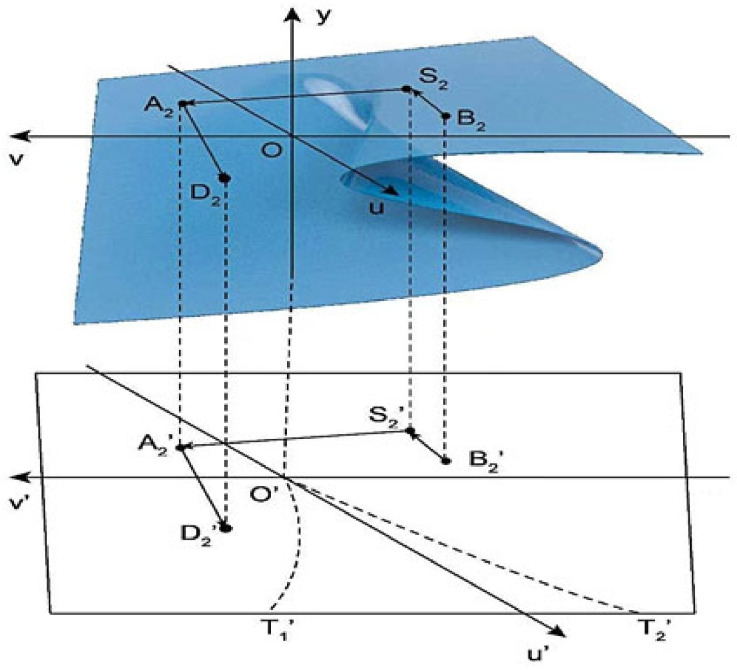
Evolution mechanism of safety state in cusp catastrophe model.

**Figure 3 ijerph-19-16744-f003:**
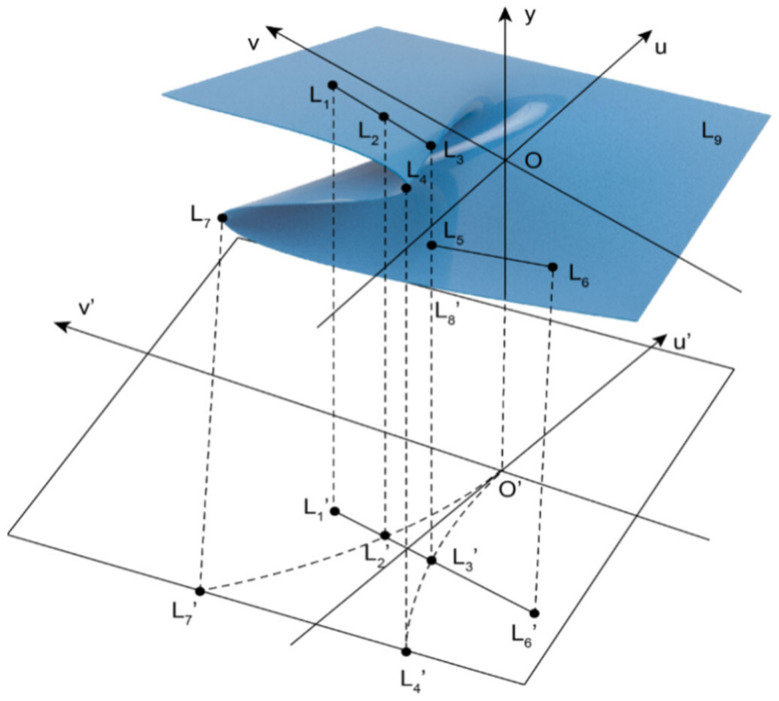
Evolution mechanism of risk state in cusp catastrophe model.

**Figure 4 ijerph-19-16744-f004:**
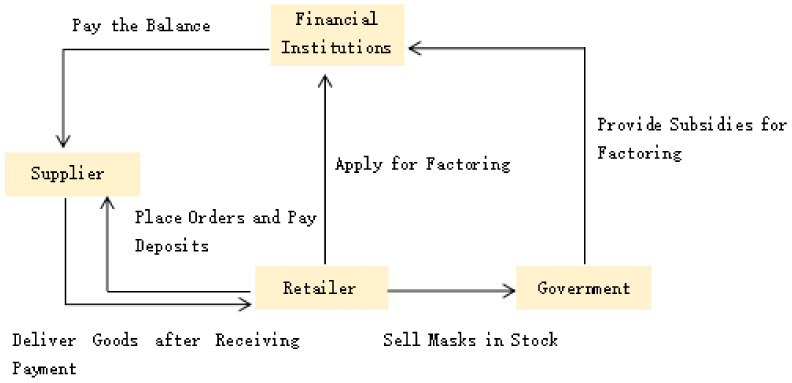
Financing model of the green supply chain for masks based on factoring.

**Figure 5 ijerph-19-16744-f005:**
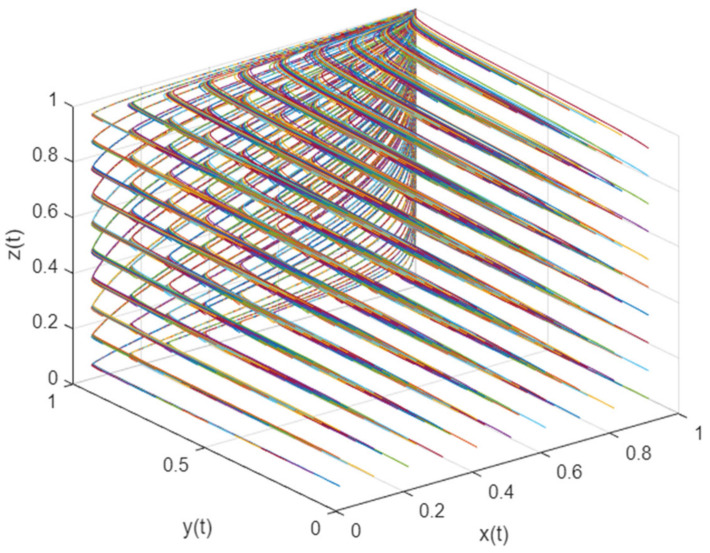
Parameter running track in the tripartite game model.

**Figure 6 ijerph-19-16744-f006:**
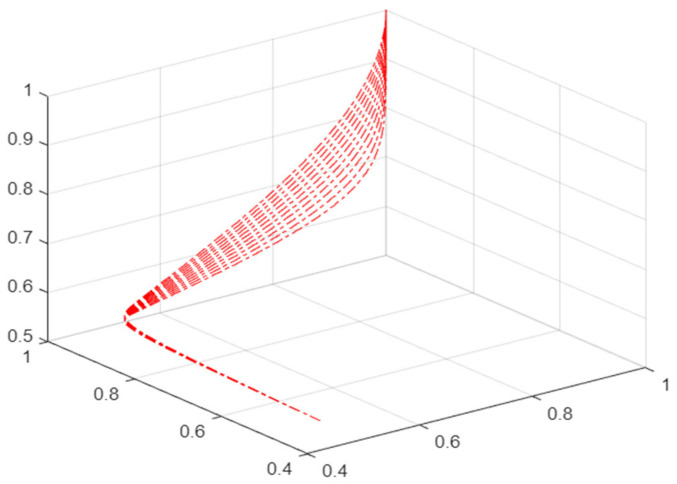
Sensitivity analysis of parameter R.

**Figure 7 ijerph-19-16744-f007:**
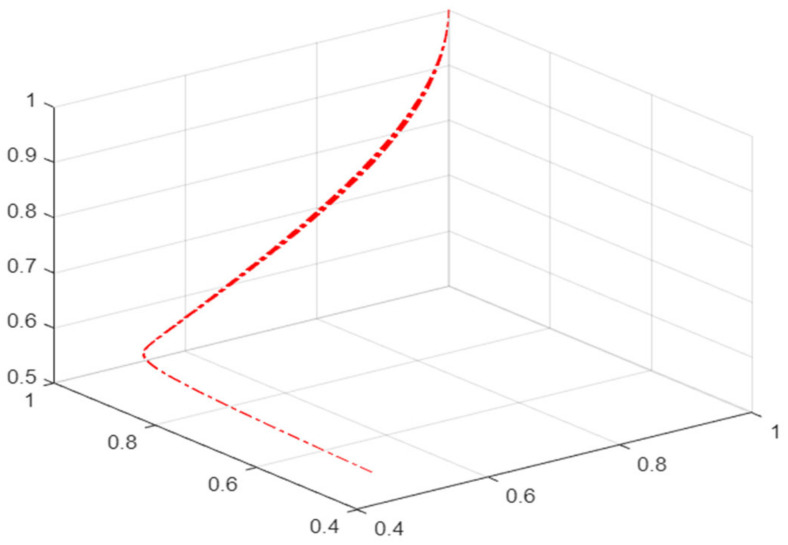
Sensitivity analysis of parameter U.

**Figure 8 ijerph-19-16744-f008:**
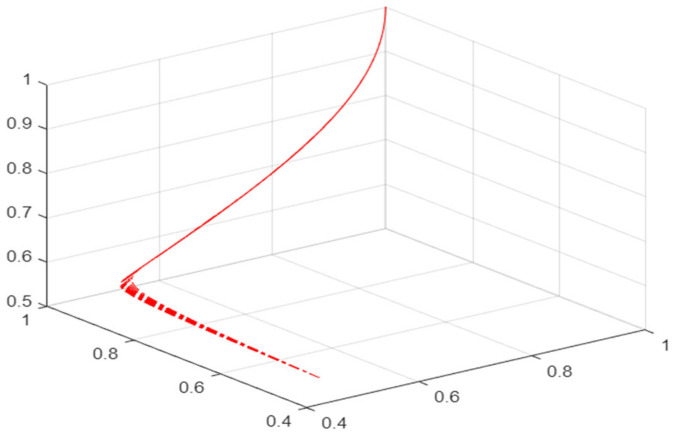
Sensitivity analysis of parameter E.

**Figure 9 ijerph-19-16744-f009:**
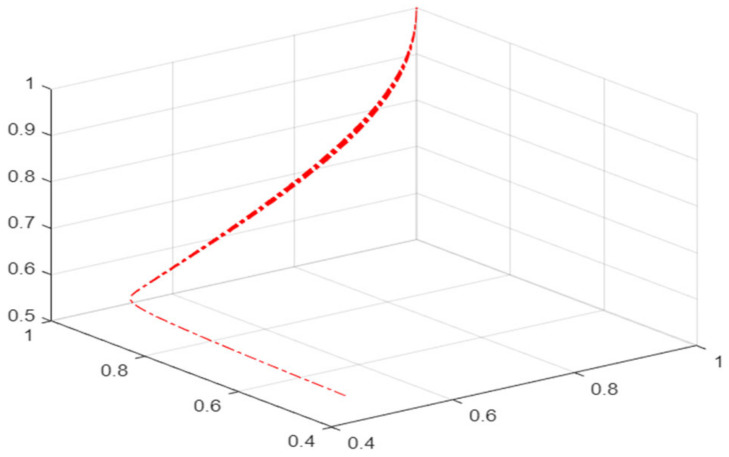
Sensitivity analysis of parameter F2.

**Figure 10 ijerph-19-16744-f010:**
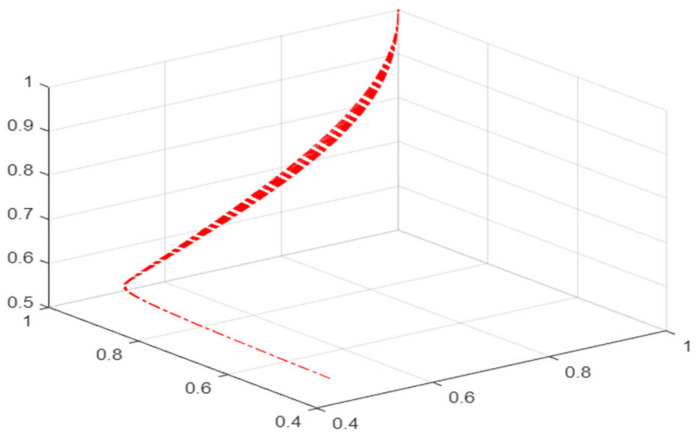
Sensitivity analysis of parameter t.

**Table 1 ijerph-19-16744-t001:** Revenue Matrix of Two-party Game Model.

Strategy Combination	Mask Supplier	Mask Retailer
(1,1)	mn(P_2_ − P_1_) − δm(t − 1)P_4_ + δmβ(t − 1)P_1_ − δ(1 − ε)m (n − t + 1)P_4_ + (1 − ε)δβ m(n − t + 1)P_1_ − nC_1_	m(t − 1)(P_3_ − P_2_) + m(1 − ε) (n − t + 1)(P_3_ − P_2_) + δm(t − 1)P_4_ + δ(1 − ε)m(n − t + 1)P_4_ − (n − t + 1)C_2_ − nC_3_
(1,0)	m(t − 1)(P_2_ − P_1_) − δm(t − 1)P_4_ + δmβ(t − 1)P_1_ + λmP_2_ − mP_1_ − (t − 1)C_1_ + D	m(t − 1)(P_3_ − P_2_) − λmP_2_ + δm(t − 1)P_4_ + (1 − ε)mP_3_ − C_2_ − (t − 1)C_3_ − D
(0,1)	mn(P_2_ − P_1_) − δm(t − 1)P_4_ + δmβ(t − 1)P_1_ − δ(1 − ε)m (n − t + 1)P_4_ + (1 − ε)δβ m(n − t + 1)P_1_ − nC_1_	m(t − 1)(P_3_ − P_2_) + m(1 − ε) (n − t + 1)(P_3_ − P_2_) + δm(t − 1)P_4_ + δ(1 − ε)m(n − t + 1)P_4_ − (n − t + 1)C_2_ − n(C_3_ + C_4_)
(0,0)	mt(P_2_ − P_1_) − δm(t − 1)P_4_ + δmβ(t − 1)P_1_ − (t − 1)C_1_ + D	m(t − 1)(P_3_ − P_2_)-mP_2_ + (1 − ε)mP_3_ +δm(t − 1)P_4_ − C_2_ − (t − 1)C_3_ − tC_4_ − D

**Table 2 ijerph-19-16744-t002:** Revenue Matrix of Tripartite Game Model.

Strategy Combination	Financial Institutions	Mask Retailer	Government
(1,1,1)	$V\left(1-\lambda \right){\left(1+r\right)}^{n}-V\left(1-\lambda \right)-W_{1}-W_{2}+F_{1}$	$V\left(1-U\right)\left(1+E\right)-V\left(1-\lambda \right){\left(1+r\right)}^{n}-V\lambda +F_{2}+\mathrm{VU}$	$R-\mathrm{VU}-F_{1}-F_{2}$
(1,0,1)	$V\left(1-\lambda \right){\left(1+r\right)}^{n-t}-V\left(1-\lambda \right)-W_{1}-W_{2}+F_{1}-W_{3}-\left(1-p\right)V\left(1-\lambda \right)+p\left[V\left(1-\lambda \right)+W_{4}\right]+\mathrm{VU}$	$V\left(1-U\right)\left(1+E\right)-V\left(1-\lambda \right){\left(1+r\right)}^{n-t}-V\lambda +\left(1-p\right)V\left(1-\lambda \right)-p\left[V\left(1-\lambda \right)+W_{4}\right]$	$R-\mathrm{VU}-F_{1}$
(1,1,0)	$V\left(1-\lambda \right){\left(1+r\right)}^{n}-V\left(1-\lambda \right)-W_{1}-W_{2}$	$V\left(1-U\right)\left(1+E\right)-V\left(1-\lambda \right){\left(1+r\right)}^{n}-V\lambda$	*0*
(0,1,1)	$V\left(1-\lambda \right){\left(1+r\right)}^{n-t}-V\left(1-\lambda \right)-W_{1}-W_{2}-Y$	$V\left(1-U\right)\left(1+E\right)-V\left(1-\lambda \right){\left(1+r\right)}^{n-t}-V\lambda +F_{2}+\mathrm{VU}+Y$	$R-\mathrm{VU}-F_{2}$
(0,0,1)	$V\left(1-\lambda \right){\left(1+r\right)}^{n-t}-v\left(1-\lambda \right)-W_{1}-W_{2}$	$V\left(1-U\right)\left(1+E\right)-V\left(1-\lambda \right){\left(1+r\right)}^{n-t}-V\lambda +\mathrm{VU}$	R−VU
(0,1,0)	$V\left(1-\lambda \right){\left(1+r\right)}^{n-t}-V\left(1-\lambda \right)-W_{1}-W_{2}-Y$	$V\left(1-U\right)\left(1+E\right)-V\left(1-\lambda \right){\left(1+r\right)}^{n-t}-V\lambda +Y$	0
(1,0,0)	$V\left(1-\lambda \right){\left(1+r\right)}^{n-t}-V\left(1-\lambda \right)-W_{1}-W_{2}-W_{3}-\left(1-p\right)V\left(1-\lambda \right)+p\left[V\left(1-\lambda \right)+W_{4}\right]$	$V\left(1-U\right)\left(1+E\right)-V\left(1-\lambda \right){\left(1+r\right)}^{n-t}-V\lambda +\left(1-p\right)V\left(1-\lambda \right)-p\left[V\left(1-\lambda \right)+W_{4}\right]$	0
(0,0,0)	$V\left(1-\lambda \right){\left(1+r\right)}^{n-t}-V\left(1-\lambda \right)-W_{1}-W_{2}$	$V\left(1-U\right)\left(1+E\right)-V\left(1-\lambda \right){\left(1+r\right)}^{n-t}-V\lambda$	0

## Data Availability

Data are contained within the article. The data presented in this study are available in Research on the anti-risk mechanism of mask green supply chain from the perspective of cooperation between retailers, suppliers, and financial institutions.

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
