# Peer review of "Research on the Anti-Risk Mechanism of Mask Green Supply Chain from the Perspective of Cooperation between Retailers, Suppliers, and Financial Institutions"

_ijerph, 2022, doi:10.3390/ijerph192416744_

Round 1

Reviewer 1 Report

-The author studied “ Research on the anti-risk mechanism of mask green supply 2 chain from the perspective of cooperation between retailers, 3 suppliers, and financial institutions”.

-As a result of the well-composed, logical structure of the abstract, the reader can easily understand the purpose of the research. The aim of the research is clear, the methodology is well-detailed.

-The abstract is very long, a maximum of 200 words should be written, the longer discussion should be discussed in the introduction.

-There are no references in any chapter of the text. The author has listed a number of references at the end of the article, but these should be numbered in the text.

-After the tables, you might want to leave out a minimum amount of space, because the table merges with the text.

-Are the formulas and figures created by the author? In many cases, there is no reference to the formulas and figures. The formulas seem to be correct.

-The order of the references in the text is correct.

-The form of the references does not follow the template. Please ask the author to correct the format according to the template.

-The article is grammatically correct and contains no typos.

-The size and quality of the illustrations are good and easy to read. However, some figures are centered, not right-aligned.

-Overall, the study is of acceptable quality, supporting the claims of the author. The length of the article is also adequate.

Author Response

        Thank you for your letter and for the reviewers’ comments concerning our manuscript entitled“Research on the anti-risk mechanism of mask green supply chain from the perspective of cooperation between retailers, suppliers, and financial institutions” (ID: IJERPH-2083390).These comments are all valuable and very helpful for revising and improving our paper, and provide crucial guidance for our research. We have studied the comments carefully and have made corrections that we hope will meet with your approval. The main corrections in the paper and the responses to the reviewers’ comments are as follows:

Comment 1:

The abstract is very long, a maximum of 200 words should be written, the longer discussion should be discussed in the introduction.

Response:

Thank you for your comments and suggestions.We have reduced the abstract to 200 words.Please see the abstract in the paper.

Comment 2:

There are no references in any chapter of the text. The author has listed a number of references at the end of the article, but these should be numbered in the text.

Response:

Thank you for your comments and suggestions.We have numbered the references in the text according to your requirements, and we have revised the references numbered in the text according to the format of the IJERPH.Please see the paper.

Comment 3:

After the tables, you might want to leave out a minimum amount of space, because the table merges with the text.

Response:

Thank you for your comments and suggestions.We have left a space below the table according to your request.Please see the line 281 and the line 471.

Comment 4:

Are the formulas and figures created by the author? In many cases, there is no reference to the formulas and figures. The formulas seem to be correct.

Response:

Thank you for your comments and suggestions.The formula derived in lines 333 –342 in this paper is to convert the copied dynamic equation of the two-sided game model into the balance surface equation of the sharp-point mutation model, The derivation of these formulas is the reference [46]: " Jiang, F.,Hu, B.,2019. Random catastrophe analysis of evolution of labor disputes behavior and stability. Journal of System Managment. 28, 991–997."The source literature has been marked in the original text.Formula derivation of the same type has been applied in another published paper "Chen, H., Wang, Z., Yu, X., 2021. Sustainability strategies of equipment introduction and overcapacity risk sharing in mask emergence supply chains during pandemics. Sustainability. 13, 1–17".Figures 1 to 3 in this paper are redrawn with reference to the cusp catastrophe model commonly seen in published papers. Figure 5 - Figure 10 is generated by using the open source matlab drawing code.

Comment 5:

The form of the references does not follow the template. Please ask the author to correct the format according to the template.

Response:

Thank you for your comments and suggestions.We have corrected the format according to the template.Please see the red part from line 588-685.

Comment 6:

The size and quality of the illustrations are good and easy to read. However, some figures are centered, not right-aligned.

Response:

Thank you for your comments and suggestions.We have aligned the related figures to the right.Please see the Figure 4.

Reviewer 2 Report

The topic of the paper is of theoretical and practical significance.

The paper is qualified for the publication after a minor revision as follows:

1.Some figures such as Figure1-3 can be more concise and tighter.

2.Some equations need to be modified to a good looking form, i.e. equation (5), and formula from line 477-490 using equation editing.

Author Response

Thank you for your letter and for the reviewers’ comments concerning our manuscript entitled“Research on the anti-risk mechanism of mask green supply chain from the perspective of cooperation between retailers, suppliers, and financial institutions” (ID: IJERPH-2083390).These comments are all valuable and very helpful for revising and improving our paper, and provide crucial guidance for our research. We have studied the comments carefully and have made corrections that we hope will meet with your approval. The main corrections in the paper and the responses to the reviewers’ comments are as follows:

Comment 1:

Some figures such as Figure1-3 can be more concise and tighter.

Response:

Thank you for your comments and suggestions.We have narrowed the relevant Figures. Please see the Figure1-10 in the paper.

Comment 2:

Some equations need to be modified to a good looking form, i.e. equation (5), and formula from line 477-490 using equation editing.

Response:

Thank you for your comments and suggestions.We have modified the equations and formula.Please see the equation (5)  and the red part from line 477-478.
